# Medication safety with oral antitumour therapeutics in paediatrics (youngAMBORA): A mixed-methods approach towards a tailored care program

Phyllis Lensker[1,2,3,4], Lisa Cuba[1,2,3], Katja Gessner[2,3], Martin F. Fromm[2,3,5], Frank Dörje[1,3,5], Markus Metzler[3,4] *

1 Pharmacy Department, Universitätsklinikum Erlangen and Friedrich-Alexander-Universität Erlangen-Nürnberg, Erlangen, Germany, 2 Institute of Experimental and Clinical Pharmacology and Toxicology, Friedrich- Alexander-Universität Erlangen-Nürnberg, Erlangen, Germany, 3 Comprehensive Cancer Center Erlangen-EMN, Universitätsklinikum Erlangen, Erlangen, Germany, 4 Department of Paediatrics and Adolescent Medicine, Universitätsklinikum Erlangen and Friedrich-Alexander-Universität Erlangen-Nürnberg, Erlangen, Germany, 5 FAU NeW–Research Center New Bioactive Compounds, Friedrich-Alexander-Universität Erlangen-Nürnberg, Erlangen, Germany

* markus.metzler@uk-erlangen.de

**Data Availability Statement:** All relevant data are already included in the article and its supporting information files.

## Abstract

### Objective

Oral antitumour therapeutics (OAT) are increasingly used due to improvements in outcomes and their convenient application. However, complex intake regimens pose several challenges. The randomised AMBORA trial (Medication Safety With Oral Antitumour Drugs) demonstrated highly positive outcomes of a clinical pharmacological/pharmaceutical care program for adults treated with numerous OAT, but comparable concepts in paediatrics are lacking so far.

### Methods

We used a parallel mixed-methods approach to develop a tailored pharmacological/pharmaceutical care program for OAT in paediatrics (youngAMBORA). We combined a quantitative analysis of tumour entities and used OAT in a paediatric cancer centre with a qualitative survey for patients, caregivers, and healthcare professionals to identify particular demands and educational needs (e.g., application problems, side effects).

### Results

Leukaemia (77/315) and antimetabolites (95/151) were the most frequently observed entity and OAT, respectively. Of 22 surveyed patients, 81.8% wanted to be involved in oral medication education. Compared to caregivers, significantly more healthcare professionals graded the three most common application problems to be challenging ('*Smell/taste*': 32/36 vs. 23/42, $p = 0.001$; '*Refusal of intake*': 31/36 vs. 16/42, $p<0.001$; '*Swallowing problems*': 28/36 vs. 21/42, $p = 0.011$). We identified nine relevant side effects, of which two (*'Skin dryness'*, *'Taste changes'*) were not included in 15 previously published core side effects of the Common Terminology Criteria of Adverse Events (CTCAE) item library.

**Funding:** PL received funding for the present work. The full names of the funders are: 'Elterninitiative krebskranker Kinder e.V.' and 'Tigerauge: Initiative Kinderhospiz Nordbayern e.V.' The URL of funders' websites are: https://www.kinder-erlangen.de/ and https://www.tigerauge.org/home.html. Grant numbers are not applicable. The funders did not play any role in the study design, data collection and analysis, decision to publish, or preparation of the manuscript.

**Competing interests:** I have read the journal's policy and the authors of this manuscript have the following competing interests: Financial competing interests: KG has received lecture fees from AstraZeneca. MFF has received consultancy fees and lecture fees from Boehringer Ingelheim and third-party funds for research projects at his institution by Boehringer Ingelheim and Heidelberg Pharma Research GmbH. FD has received consultancy fees from Lilly Deutschland and SANDOZ AG and has received lecture fees from E. Lilly and Janssen. KG, MFF, and FD received an earmarked financial contribution for the first award of the MSD Germany Health Award 2021. PL, LC, and MM declared that no financial competing interests exist. Non-financial interests: The authors have declared that no non-financial competing interests exist.

## Conclusion

Based on the present findings, the tailored youngAMBORA care program will include: 1) counselling sessions for classic and targeted OAT, 2) child-friendly support with drug application, and 3) systematic evaluation of 17 relevant side effects from patients' and caregivers' points of view including age-appropriate information material.

## Introduction

The increasing use of genetic testing and an improved understanding of molecular signatures directed the treatment of many adult and paediatric malignancies towards targeted therapies [1–3]. Therefore, cancer treatment has been substantially impacted by the clinical use of oral antitumour therapeutics (OAT) over the past two decades [4]. On the one hand, increased convenience without the need for intravenous access is associated with improved quality of life for both paediatric and adult patients [5–7]. On the other hand, potential safety and efficacy concerns arise from the shift of responsibility towards the patients and caregivers due to difficult handling instructions, complex intake regimens, or drug-drug/drug-food interactions [8].

Parents of children affected by malignant diseases typically experience intense psychological distress [9]. Apart from the burdens related to the diagnosis, parents must acquire the necessary knowledge about medical care and medication handling in the home setting [10]. Previous studies showed, that parents of children receiving OAT stated a lack of knowledge about proper handling and difficulties in drug administration [11]. Beyond that, many drugs prescribed for paediatric oncological patients are used without approval in the palliative setting and are not commercially available as age-appropriate formulations [12, 13]. These challenges require tailored support for patients and their caregivers.

The randomised multicentre AMBORA trial [14] (Medication Safety With Oral Antitumour Drugs; 11/2017-01/2020) demonstrated highly positive outcomes of an intensified clinical pharmacological/pharmaceutical care program for adult patients treated with a broad variety of newer OAT. Within the first twelve weeks of oral anticancer treatment, this program led to a significant reduction of drug-related problems (e.g., unresolved medication errors and side effects) by 34% in the intervention group. Importantly, the probability of reaching a combined endpoint of severe side effects, treatment discontinuation, unscheduled hospitalisation, or death was significantly reduced by 52% in the intervention group [14].

Similar care programs for paediatric patients treated with OAT are lacking. We therefore set out to develop a tailored pharmacological/pharmaceutical care program (youngAMBORA) for children and adolescents. By combining quantitative and qualitative data collection within a parallel mixed-methods approach, we aimed to 1) characterise the targeted patient population and compare it with the adult AMBORA trial cohort, 2) identify the educational needs in children, caregivers, and healthcare professionals and 3) adapt the evidence-based AMBORA intervention to meet the challenging requirements in paediatric oncology.

## Materials and methods

As the quantitative part of the mixed-methods approach, we retrospectively evaluated tumour entities and frequently used OAT of patients treated in the paediatric cancer centre to characterise and compare the paediatric and adult patient populations (Fig 1). As the qualitative part, we conducted a prospective survey for patients, caregivers, and healthcare professionals with

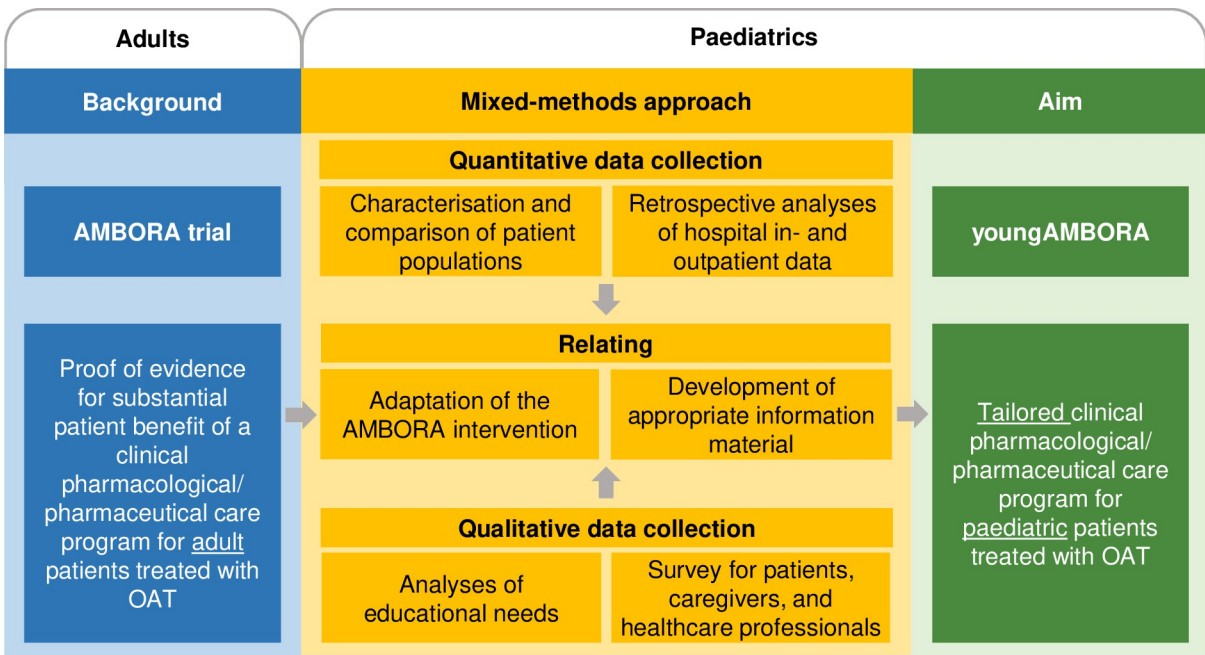

**Fig 1. Mixed-methods approach to develop a tailored care program for paediatric patients treated with OAT (youngAMBORA).**
Abbreviations: AMBORA = Medication Safety With Oral Antitumour Drugs; OAT = Oral antitumour therapeutics.

the aim of identifying contents of structured counselling sessions addressing the educational needs (Fig 1). By combining the findings of both evaluations, we developed targeted information material and adapted the adult AMBORA intervention to the paediatric setting (Fig 1).

The present study was conducted at the University paediatric cancer centre, which is part of the Comprehensive Cancer Centre Erlangen-EMN and includes an inpatient ward, a stem cell transplantation unit, a day-clinic, an outpatient palliative care team, and an outpatient-clinic.

## Characterisation and comparison of patient populations

For the quantitative data collection, hospital in- and outpatient data were retrospectively analysed. This most comprehensive, recent and reliably available data set included the number of treated oncological entities between 1st January 2017 and 31st December 2021 and ambulatory prescription data of OAT between 1st October 2020 and 1st October 2022. The distribution of tumour entities and prescribed OAT were subsequently compared to the adult AMBORA trial cohort. Only retrospective, anonymised data were used, that did not allow the identification of any individual patient.

## Analyses of educational needs

For the qualitative data collection, a single time-point, paper-pencil survey for patients, caregivers, and healthcare professionals was conducted in the paediatric cancer centre until the time point at which no further changes in the results were observed (between 1st February 2023 and 31st August 2023). Tailored questionnaires were designed for each group. Questions were presented in both open-ended, single-, and multiple-choice formats. When applicable, 5-point Likert scales including a neutral response option were chosen. Child-friendly answer options (e.g., smileys) were used for the patient version and also contained a neutral feedback option.

Patients and caregivers were eligible to participate in the survey, if they/their child were either receiving at least one oral non-oncological drug or OAT at the time of data collection or had been treated with OAT or any other oral drug in the past and if an oncological diagnosis was more than three months ago. According to the inclusion criteria for other validated questionnaires commonly used in paediatrics [15, 16], patients from the age of six years and healthcare professionals (nurses and physicians) working in the paediatric cancer centre were included in the survey.

Based on the principles of patient and public involvement in research [17], all three versions of questionnaires were peer-reviewed, discussed, and piloted by two representatives of each group as well as by clinical pharmacologists/pharmacists experienced in oncology. Questionnaires were returned in labelled boxes in all treatment units and data was recorded anonymously.

The survey comprised three main sections: *Section A 'Administration and handling of oral drugs'* to identify challenges associated with oral drug application in general, *section B 'Oral medication education'* to discern existing gaps, and *section C 'Side effects'* of anticancer treatment to determine the relevance of side effects evaluated from the three survey groups' respective points of view. We inquired 62 side effects based on the paediatric module of the patient-reported outcomes version of the Common Terminology Criteria of Adverse Events (Ped-PRO-CTCAE®) measurement system [18]. Furthermore, the survey groups' classification (relevant, neutral, or less relevant) for the respective side effects was compared with 15 core CTCAE terms previously established by Reeve et al. [19].

## Adaptation of the AMBORA intervention

Elements of the AMBORA intervention are described in the supporting methods. Based on the findings of the quantitative and qualitative data collection, these elements were adapted to the paediatric setting. Tailored information material for paediatrics was designed in consideration of 1) existing material from the AMBORA trial, 2) clinical characteristics of paediatric oncology patients, and 3) educational needs identified in the survey. Drug fact sheets were created for the most commonly prescribed OAT identified in the quantitative data analysis. Brochures about side effects were created for the most relevant side effects identified within the qualitative evaluation. All information material was created based on standard operating procedures and double-checked by two clinical pharmacologists/pharmacists.

## Statistical analysis

Data was analysed using descriptive statistics and differences were regarded as significant when *p* values were <0.05 (two-sided chi-square test or Wilcoxon signed rank test). Analyses were performed using Microsoft Excel and GraphPad Prism Version 9. Incomplete surveys were included, but responses were analysed based on the number of complete responses for each question.

## Ethics approval

The present investigation is in accordance with the Bavarian Hospital Act (Bayerisches Krankenhausgesetz, BayKrG) Article 27(4). On the same basis, the Ethics Committee of the Friedrich-Alexander-Universität Erlangen-Nürnberg provided a waiver (23–273_1-ANF) for data collection within the youngAMBORA care program. The need for informed consent was waived in consideration of the Bavarian Hospital Act Article 27(4).

## Results

### Characterisation and comparison of patient populations

From 01/2017 to 12/2021, 315 patients were treated in the paediatric cancer centre (Fig 2A). Distribution of tumour entities was representative for the spectrum of childhood malignancies in Germany [20] and included leukaemia (24.4%; 77), tumours of the central nervous system (21.9%; 69), and lymphoma (17.1%; 54; Fig 2A). Entities that were included in the AMBORA trial were characteristic for malignant diseases in adults, whose standard treatment includes oral antitumour therapy [breast cancer (17.8%; 36), neuroendocrine tumours (9.9%; 20) and prostate cancer (9.9%; 20); Fig 2B].

A total of 151 OAT (composed of 26 different drugs) was newly prescribed in the paediatric cancer centre between 10/2020 and 10/2022 (Fig 3A). In the AMBORA trial, 202 patients with

**A**

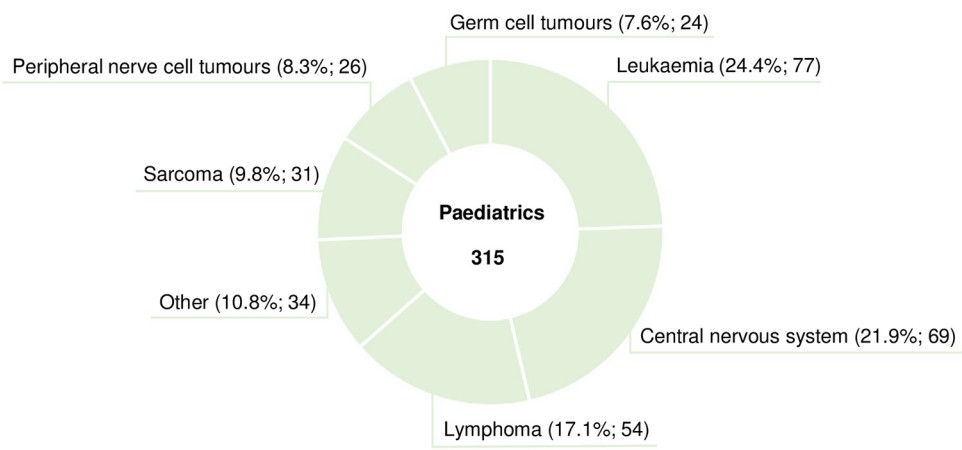

**B**

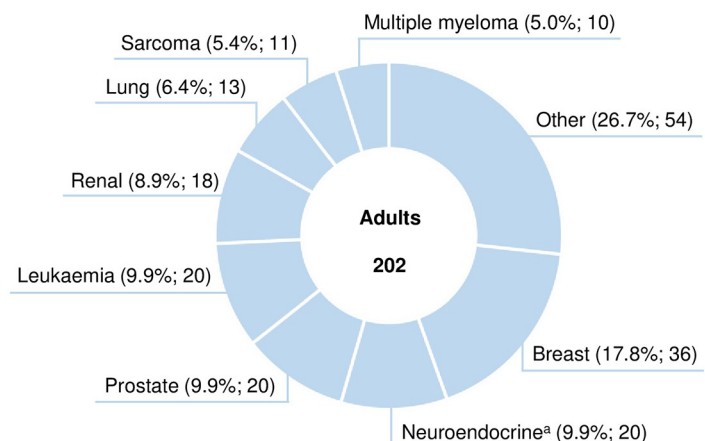

**Fig 2. Quantitative analysis of tumour entities in the paediatric cancer centre and the AMBORA trial.** (A) Distribution of oncological entities in the paediatric cancer centre; 315 = Entirety of patients treated in the paediatric cancer centre between 01/2017 and 12/2021, (B) Distribution of oncological entities in the AMBORA trial (adult patients) [14]; 202 = Entirety of patients included in the AMBORA trial. [a]Neuroendocrine tumours of gastrointestinal, pulmonal, renal, or pancreatic origin.

A

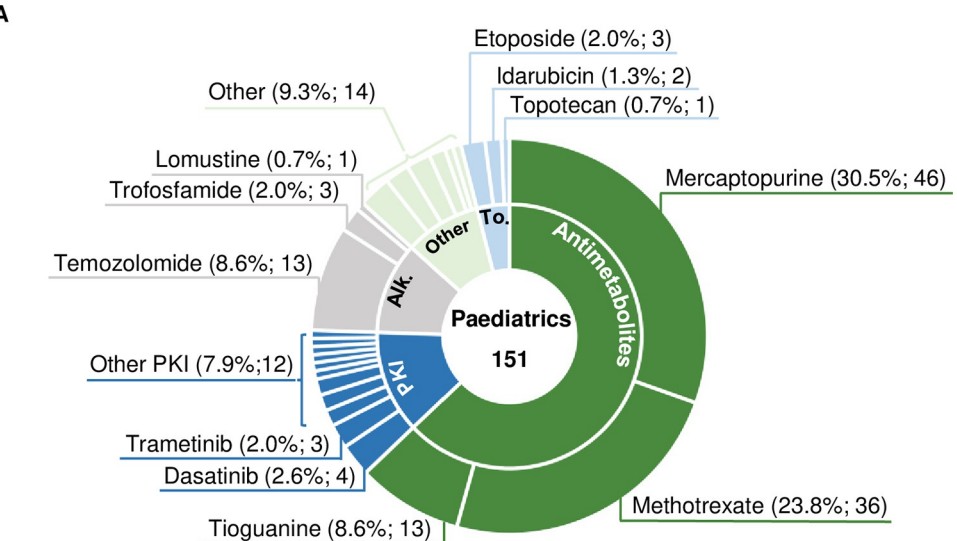

B

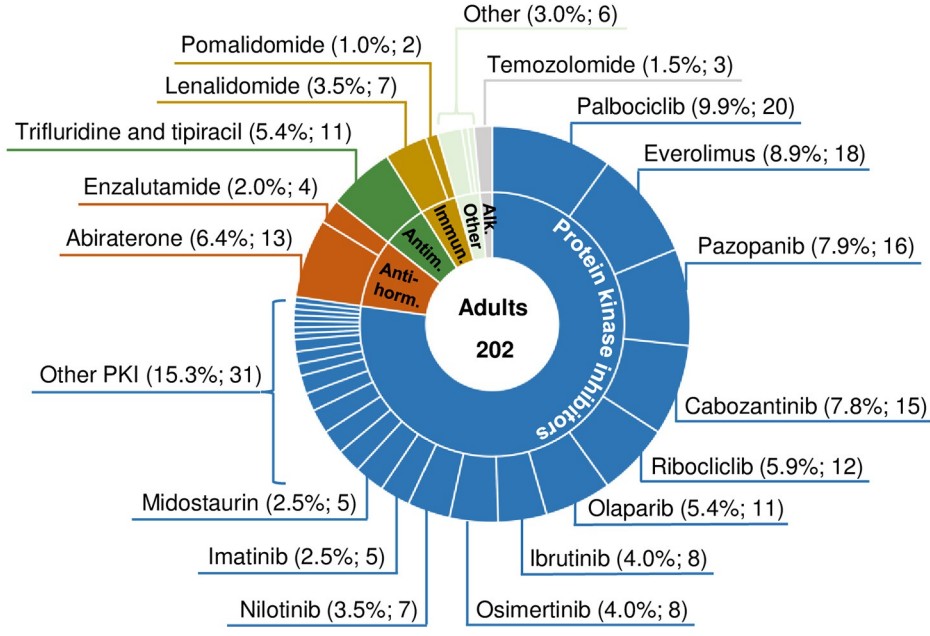

**Fig 3. Quantitative analysis of oral antitumour therapeutics in the paediatric cancer centre and the AMBORA trial.** (A) Used oral antitumour therapeutics in the paediatric cancer centre between 10/2020 and 10/2022; 151 = Entirety of first prescriptions of oral antitumour therapeutics between 10/2020 and 10/2022, (B) Used oral antitumour therapeutics in the AMBORA trial (adult patients) [14]; 202 = Entirety of patients included in the AMBORA trial. Abbreviations: PKI = Protein kinase inhibitors; Alk. = Akylants; To. = Topoisomerase inhibitors; Anti-horm. = Anti-hormones; Antim. = Antimetabolites; Immun. = Immunomodulators.

35 different OAT were included (Fig 3B). Whereas protein kinase inhibitors represented 77.2% (156/202) of all prescribed OAT in the adult cohort, they only accounted for 12.6% (19/151) in the paediatric setting. Antimetabolites were the most frequently used OAT in paediatrics [62.9%; 95/151, including mercaptopurine as the most common OAT (30.5%; 46/151)]. Of all prescribed OAT for children, only 20.5% (31/151) would have met the inclusion criteria

**Table 1. Characteristics of surveyed patients, caregivers, and healthcare professionals.**

| Patients (N = 22) | |
|---|---|
| Current or previous OAT intake by patient; Number (%) | 17 (77.3) |
| Age of patient (years); Median (range) | 9.5 (6–15) |
| *Age 6–8 years; Number (%)* | 8 (36.4) |
| *Age 9–12 years; Number (%)* | 7 (31.8) |
| *Age 13–18 years; Number (%)* | 7 (31.8) |
| **Caregivers (N = 44)** | |
| Current or previous OAT intake by patient; Number (%) | 29 (65.9) |
| Age of patient (years); Median (range) | 6.0 (1–21) |
| **Healthcare professionals (N = 36)** | |
| Profession; Number (%) | |
| *Nurse* | 19 (52.8) |
| *Physician* | 17 (47.2) |
| Working area; Number (%) | |
| *Oncology ward* | 24 (66.7) |
| *Stem cell transplantation unit* | 9 (25.0) |
| *Day-clinic* | 8 (22.2) |
| *Outpatient palliative care team* | 8 (22.2) |
| *Outpatient-clinic* | 7 (19.4) |
| Work experience (years); Median (range) | 7.0 (2–40) |

from the AMBORA trial (Fig 3) [14]. The proportion of patients treated with OAT that are not approved in the respective indications ('off-label') was significantly higher in the paediatric cohort compared to adults (19.9%; 30/151 vs. 9.4%; 19/202, $p$ = 0.005). Of the 26 different OAT prescribed in the paediatric cancer centre, 65.4% (17/26) were not approved for children.

## Analyses of educational needs

Overall, 102 surveys were completed, thereof 22 from patients, 44 from caregivers, and 36 from healthcare professionals. Characteristics of surveyed patients, caregivers, and healthcare professionals are shown in Table 1. All survey questions are provided in the supporting information (S1-S9 Tables in S1 File).

## Administration and handling of oral drugs (Section A)

Of surveyed children, 77.3% (17/22) indicated not to be afraid of taking oral drugs (S1 Table in S1 File). Nevertheless, 76.7% (33/43) of paediatric patients had not yet been confronted with the regular intake of any oral drugs before cancer diagnosis, according to their caregivers (S2 Table in S1 File).

The top three challenges associated with oral drug intake mentioned by caregivers and healthcare professionals were dealing with the application itself (Fig 4A). Significantly more healthcare professionals graded these problems to be challenging compared to the caregivers (88.9%; 32/36 vs. 54.8%; 23/42, $p<0.001$ for *'Smell/taste'*, 86.1%; 31/36 vs. 38.1%; 16/42, $p$ = 0.001 for *'Refusal of intake'*, 77.8%; 28/36 vs. 50.0%; 21/42, $p$ = 0.011 for *'Swallowing problems'*). All three groups did not consider *'Adherence'* as a demanding point (Fig 4A, S1 Table in S1 File).

According to the caregivers, swallowing problems were significantly reduced over treatment time (mean 4.0 ± standard deviation 1.4 vs. 2.3 ± 1.5, $p<0.001$; Fig 4B). While 85.7% (36/42) of caregivers preferred liquid formulations for their children at the start of anticancer treatment (Fig 4C), 47.2% (17/36) changed their evaluation and favoured solid drug forms over

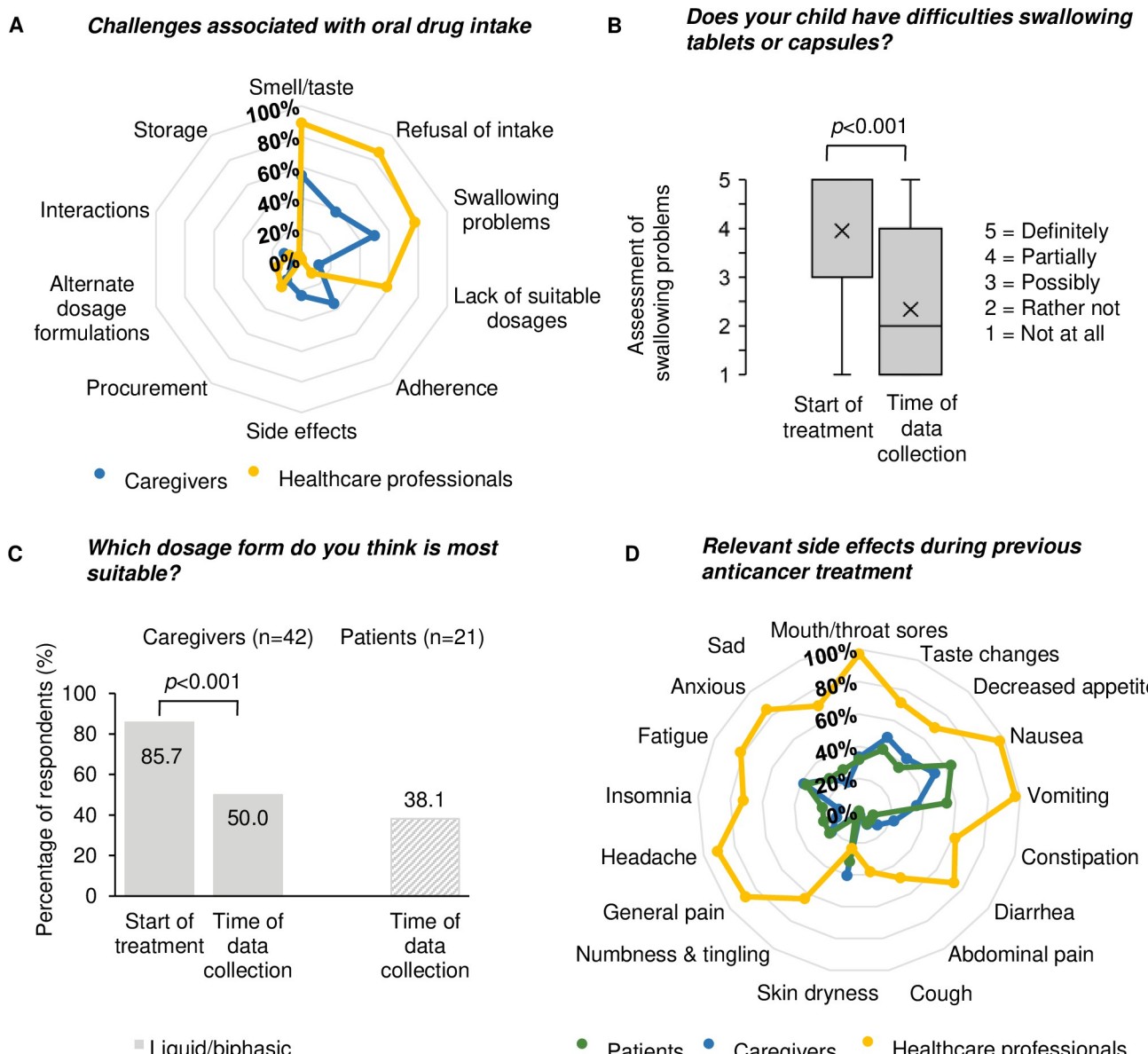

**Fig 4. Qualitative analysis of educational needs in the paediatric cancer centre: Selected survey results.** (A) Proportion of caregivers (n = 42) and healthcare professionals (N = 36) describing challenges associated with oral drug intake; (B) Assessment of swallowing difficulties mentioned by caregivers (N = 44) retrospectively for the start of treatment and for the time of data collection; data was analysed using Wilcoxon signed rank test, box plot with 25th and 75th percentiles with '+' representing the mean and horizontal line representing the median; (C) Proportion of caregivers (n = 42, single-coloured) and patients (n = 21, patterned) favouring liquid/biphasic dosage forms retrospectively for the start of treatment and for the time of data collection; (D) Proportion of patients, caregivers, and healthcare professionals describing side effects during previous anticancer treatment (oral and/or parenteral) as being relevant (core CTCAE terms published by Reeve et al. [19] + CTCAE terms mentioned as being relevant by at least 20% of all three groups in the survey).

time. In line with that, only 38.1% of surveyed patients considered liquid drug formulations as the most convenient ones at the time of data collection (8/21; Fig 4C).

## Oral medication education (Section B)

Of the surveyed patients, 81.8% (18/22) wanted to be involved in oral medication education (S4 Table in S1 File). Even though, caregivers and healthcare professionals regarded the

information about oral drugs at the time of a new prescription to be sufficient (S5 and S6 Tables in S1 File), the majority of both groups (81.4%; 35/43; S5 Table in S1 File and 91.7%; 33/36; S6 Table in S1 File) considered an additional pharmacological/pharmaceutical consultation to be useful. Prevention and handling of side effects was described as a key element by 57.7% (15/26) of caregivers (S5 Table in S1 File). Healthcare professionals described various barriers in oral medication education [e.g., communication problems, lack of appropriate information material, or questions outside of their expertise (e.g., pharmaceutical topics); S6 Table in S1 File]. Both groups stated that counselling should be conducted at the time of prescription or hospital discharge and should be assisted by individual, written information material (S5 and S6 Tables in S1 File).

### Side effects (Section C)

Out of 62 CTCAE terms, we identified nine side effects, which were graded as being relevant during anticancer treatment by at least 20% of participants in all three groups (S7 Table in S1 File). Two of these side effects *('Skin dryness' and 'Taste changes')* were not included in the 15 core CTCAE terms by Reeve et al. [19] and therefore added to be systematically evaluated within the youngAMBORA care program. All 17 side effects including their relevance assessment are shown in Fig 4D. Overall, a higher proportion of healthcare professionals assessed posed side effects as being relevant compared to caregivers and patients. 81.4% (35/43) of surveyed caregivers and 80.0% (28/35) of healthcare professionals described at least partial insecurities in recognising the severity of patients' side effects (S8 and S9 Tables in S1 File).

### Adaptation of the AMBORA intervention: the youngAMBORA care program

Compared to previous pharmacologist/pharmacist-led educational programs in paediatric oncology [21, 22], the youngAMBORA care program is a long-term support for children and caregivers. Because treatment regimens for acute lymphoblastic leukaemia (ALL) require discontinuous OAT intakes, number and timing of consultation sessions will depend on the planned treatment duration. In case of continuous OAT intake for at least twelve weeks (patient group I), counselling sessions will be conducted at the time of OAT initiation (week 0), week 4, and 12. In case of discontinuous intake for at least two weeks (patient group II), patients and their caregivers will be consulted at week 0,1, and 2. Patient/caregiver-reported outcomes (e.g., side effects) will be systematically evaluated at predefined timepoints using validated questionnaires and advanced medication reviews will be carried out within every consultation session (Fig 5).

Counselling sessions will be performed with children and their caregivers using age-appropriate information material. Identified problems with drug application and relevant side effects will be addressed within the consultations. So far, 25 OAT fact sheets and nine brochures about side effects have been designed (S11 and S12 Tables in S1 File). Further explanations are provided in the supporting information, including details on the new youngAMBORA information material (S10 Table in S1 File) and the drug fact sheet for the most commonly prescribed OAT mercaptopurine (S1 Fig in S1 File).

### Discussion

In the present work, we identified child-specific requirements of an intensified pharmacological/pharmaceutical care program for oral antitumour therapeutics (OAT) using a mixed-methods approach (Fig 1). We combined a quantitative analysis of tumour entities and

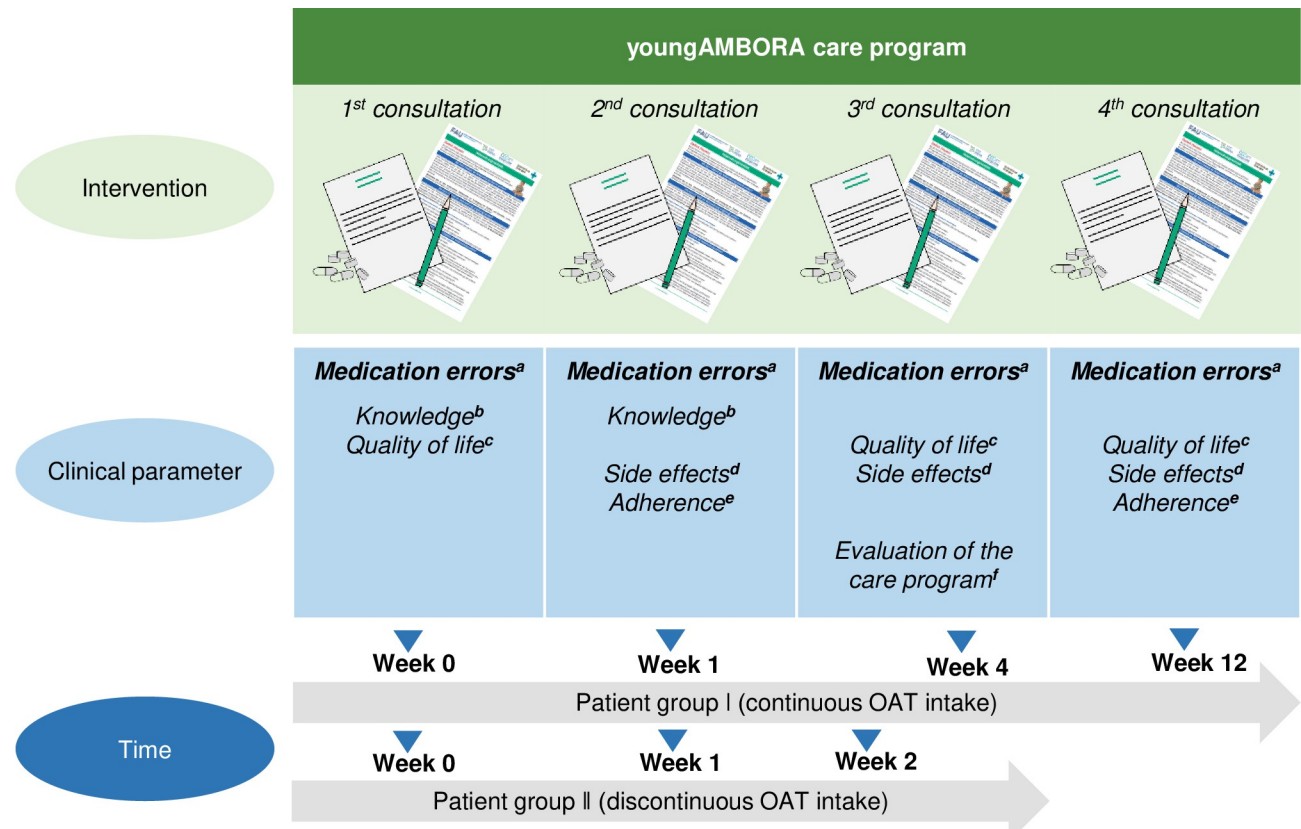

**Fig 5. The youngAMBORA care program: Pharmacological/pharmaceutical counselling sessions and measures of clinical parameters at predefined timepoints.** [a]Pharmaceutical Care Network Europe (PCNE V9.1) [23] to assess cause and National Coordinating Council for Medication Error Reporting and Prevention (NCC MERP) [24] to define severity of medication errors; [b]SIMS-D [25] to measure caregivers'/patients' knowledge; [c]Cancer module of PedsQL™ [15, 16] to record patients' health-related quality of life; [d]Ped-PRO-CTCAE® [18] to record side effects; [e]MARS-D [26] to record adherence; [f]Self-designed questionnaire to evaluate the care program. Abbreviations: OAT = Oral antitumour therapeutics.

frequently used OAT in a paediatric cancer centre with a qualitative survey for patients, caregivers, and healthcare professionals to identify qualitative educational needs.

First-line treatment of ALL, as the most frequent paediatric cancer (Fig 2A), includes oral maintenance chemotherapy with mercaptopurine and methotrexate [27], explaining the contribution of classic OAT in our evaluation (Fig 3A). Excluding paediatric patients treated with OAT approved before 2001 in line with the AMBORA trial [14] would therefore result in the majority of patients and caregivers not being reached by the youngAMBORA care program. Targeted therapies accounted for a small proportion of OAT, as they are often limited to late treatment lines or clinical trials [28, 29]. The frequently observed off-label use is a known difficulty [13] and can be explained by the shortage of clinical studies involving children and adolescents, especially in the palliative treatment setting. Palliative treatment is often perceived as the most demanding, individual situation on the one hand, but also characterised by higher healthcare capacities allowing a closer interaction between patients, caregivers, and healthcare professionals on the other hand. The associated lack of clinical experience and appropriate information material contributes in particular to the need for intensified support of children treated with targeted OAT. As a result, patients and caregivers will be consulted in the youngAMBORA care program regardless of whether the treatment intention is palliative or curative.

Herziger et al. described a high proportion of children and caregivers who experienced drug-handling problems outside of the field of oncology [30]. The wide use of classic OAT bears particular risks in oral drug handling due to their carcinogenic, mutagenic, or reproductive toxic properties. The importance of swallowing difficulties as the main handling challenge (Fig 4A) is confirmed by a recent review, showing that only 11% (9/85) package inserts of FDA-approved OAT included information on alternative administration routes [12]. Despite the lack of information, pharmacological/pharmaceutical interventions including practical training and information presented as written or video instructions could previously improve safety aspects in OAT handling, if extemporaneous compounding was unavoidable [31]. Since we noticed an improvement in swallowing difficulties and a change of preference towards solid drug forms during treatment progress (Fig 4C), tailored counselling sessions at OAT initiation appear to be most promising. Similarly, pharmaceutical counselling at start of an OAT could significantly reduce knowledge deficits in a pilot study by Zimmer et al. [21].

Interestingly, 'Adherence' was not considered as a possible barrier to oral drug intake in our survey and might be underestimated by the three surveyed groups, even though mentioned administration challenges can negatively influence treatment compliance. In previous studies, the prevalence of nonadherence to oral paediatric ALL maintenance chemotherapy ranged up to 77% and was associated with poor outcomes, including risk of cancer relapse [32–34]. Effective child-pharmacist communication is generally considered as a facilitator for adherence to drug therapy [35, 36], which confirms and emphasises the wish we identified among patients for direct involvement in consultations (S4 Table in S1 File). Similar to our findings, Nordenmalm et al. [37] described the wish of children to be provided with age-appropriate medical information [37]. Since pharmacists' verbal consultation of children supported by visual education material is known to be more effective in providing information on correct drug use than providing package inserts or educational sheets [11, 36–38], we designed child-friendly information material and video clips (S10 Table in S1 File).

Despite the fact, that caregivers and healthcare professionals rated shared information for newly prescribed oral drugs as sufficient in our survey, Oberoi et al. [39] observed medication errors in nearly 25% of children receiving OAT for ALL treatment, which were predominantly administration errors [39]. Improving communication between pharmacologists/pharmacists and parents has been proven to be an effective strategy to reduce medication errors in paediatric patient care [22], which is why consultation sessions will be regularly complemented by advanced medication reviews (Fig 5).

In the context of challenges with oral drug intake, side effects appeared to be of minor relevance (Fig 4A). However, looking at the oncological therapy in general, side effects were frequently mentioned by the three surveyed groups (Fig 4D). In contrast to Freyer et al. [40], we observed an overestimation of side effects' relevance of healthcare professionals compared to patients and caregivers. As a potential explanation, we consider their broad evaluation of all previously treated patients, whereas patients and caregivers only assessed their personal experiences. The relevance of the two CTCAE terms 'Skin dryness' and 'Taste changes' might have been underestimated by Reeve et al. [19], as all three groups assessed them as being relevant in our survey.

Taking into account that requirements for an intensified OAT-care program vary from adult to paediatric cancer patients, we regard our investigation as fundamental. As other studies investigating paediatric palliative care programs showed, the pre-assessment of child- and caregiver-specific needs is crucial [41, 42]. Identified challenges can be directly addressed in the tailored youngAMBORA care program to improve OAT handling, treatment adherence, self-management of side effects, and ultimately patient outcomes. To the best of our knowledge, similar concepts to the youngAMBORA care program including a comprehensive and

continuous support of paediatric patients treated with OAT are lacking. In general, clinical pharmacologist/pharmacist services in outpatient units are not common practice within the German healthcare setting. Therefore, tailored support can lead to increased treatment safety and effectiveness, especially in the outpatient setting, which is known for a high risk of drug-related problems [43].

We consider the combination of quantitative and qualitative data collection as a major strength of our work. Although, intake of any oral drug (not only OAT) was assessed in the survey, our findings of the qualitative data analyses are a representative starting point for evolving a care program focused on anticancer drugs, as the majority of participating patients and caregivers were already experienced in OAT handling (Table 1). Beyond that, we acknowledge the involvement of paediatric patients in the survey as a notable advantage, since the children's perspective of on oral drug therapy is rarely reported [30].

There are some limitations to be noted. First of all, our evaluation was a monocentric approach with limited survey participation rates, which might reduce generalisability. The participation of more motivated patients of different ages from six onwards, caregivers, or healthcare professionals might have led to biased results of the qualitative analyses. In general, mixed-methods research is associated with the challenge of comparing and combining quantitative and qualitative data. For instance, we used the most recent and reliably accessible data for the quantitative analyses, but data interpretation might be limited by considering a different time interval in the qualitative evaluation.

Moreover, the categorisation of relevant side effects was conducted retrospectively for the prior cancer therapy, irrespective of whether it was administered orally or intravenously, which could potentially bias our results. However, this might support the suitability of our developed care program, since OAT are frequently combined with intravenous treatment.

## Conclusions

In conclusion, our mixed-methods approach led to three main findings that will be addressed in the newly developed, tailored youngAMBORA care program: 1) pharmacological/pharmaceutical counselling sessions focusing on classic and targeted OAT, 2) child-friendly support with oral drug application, and 3) systematic evaluation of 17 identified relevant side effects from patients' and caregivers' point of view including age-appropriate information regarding their prevention and management.

## Supporting information

**S1 File. Supporting information (S1-S12 Tables, S1 Fig).**
(DOCX)

## Acknowledgments

We gratefully acknowledge the healthcare professionals of the paediatric cancer centre of the Comprehensive Cancer Centre Erlangen-EMN and particularly all participating patients and caregivers. The present work was performed in (partial) fulfilment of the requirements for obtaining the degree 'Dr. rer. biol. hum.' from the Friedrich-Alexander-Universität Erlangen-Nürnberg (Phyllis Lensker).

## Author Contributions

**Conceptualization:** Phyllis Lensker, Lisa Cuba, Katja Gessner, Martin F. Fromm, Frank Dörje, Markus Metzler.

**Data curation:** Phyllis Lensker, Lisa Cuba, Katja Gessner.

**Formal analysis:** Phyllis Lensker, Lisa Cuba, Katja Gessner, Martin F. Fromm, Frank Dörje, Markus Metzler.

**Funding acquisition:** Markus Metzler.

**Investigation:** Phyllis Lensker, Lisa Cuba, Katja Gessner.

**Methodology:** Phyllis Lensker, Lisa Cuba, Katja Gessner, Martin F. Fromm, Frank Dörje, Markus Metzler.

**Project administration:** Martin F. Fromm, Frank Dörje, Markus Metzler.

**Resources:** Phyllis Lensker, Lisa Cuba, Katja Gessner.

**Supervision:** Martin F. Fromm, Frank Dörje, Markus Metzler.

**Validation:** Phyllis Lensker, Lisa Cuba, Katja Gessner.

**Visualization:** Phyllis Lensker, Lisa Cuba, Katja Gessner.

**Writing – original draft:** Phyllis Lensker, Lisa Cuba, Katja Gessner, Martin F. Fromm, Frank Dörje, Markus Metzler.

**Writing – review & editing:** Phyllis Lensker, Lisa Cuba, Katja Gessner, Martin F. Fromm, Frank Dörje, Markus Metzler.

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
