## [Decision Letter · Decision Letter 0]

10 Sep 2024

PONE-D-24-17810Medication safety with oral antitumor therapeutics in paediatrics (youngAMBORA): A mixed-methods approach towards a tailored care programPLOS ONE

Dear Dr. Metzler,

Thank you for submitting your manuscript to PLOS ONE. After careful consideration, we feel that it has merit but does not fully meet PLOS ONE’s publication criteria as it currently stands. Therefore, we invite you to submit a revised version of the manuscript that addresses the points raised during the review process. 

We look forward to receiving your revised manuscript.

Kind regards,

Sudhir Adhikari, MBBS,MD

Academic Editor

PLOS ONE

Journal Requirements:

2. In the online submission form, you indicated that all relevant data are within the manuscript and its Supporting information file. Additional individual, pseudonymized data is available upon request. 

Reviewers' comments:

Reviewer's Responses to Questions

**Comments to the Author**

1. Is the manuscript technically sound, and do the data support the conclusions?

Reviewer #1: Yes

Reviewer #2: Yes

2. Has the statistical analysis been performed appropriately and rigorously? 

Reviewer #1: Yes

Reviewer #2: Yes

3. Have the authors made all data underlying the findings in their manuscript fully available?

Reviewer #1: Yes

Reviewer #2: Yes

4. Is the manuscript presented in an intelligible fashion and written in standard English?

Reviewer #1: Yes

Reviewer #2: Yes

5. Review Comments to the Author

Reviewer #1: The sequence of events is not well timed. There is a mismatch between the collected retrospective data and questionnaire-based survey. There is sheer disconnect between sequence of events. The authors need to clarify why there is a time difference in that

Reviewer #2: 1. The study is well designed and analyzed, however, the presentation and flow of the study information could be simpler and less confusing. Authors must highlight how the study results can be clinically applicable.

2. The inclusion criteria says--if they/their child were either receiving at least one oral drug at the time of data collection or had been treated with OAT or any other oral drug in the past and if an oncological diagnosis was more than three months ago. --do the authors imply non-Oncological oral drugs also being assessed here? If that the case then the study results cannot be applied to only OAT!

3. Pediatric Oncology practice mainly involves parenteral chemo [except ALL/Ly] in the First line setting. When we use oral drugs, it is may times in the palliative chemo setting. Can authors clarify, the treatment setting for these patients. Authors should also discuss whether this could change patient/caregiver response.

4. Reliability of survey data from Kids as young as 6 years is questionable as we all know, understanding of kids can be diverse and they can be influenced easily. Kids can have inconsistence responses depending on "surroundings and incentives!" I feel if Authors can redo study by just including kids order than 8 or 9 OR have some metric of assessing kids' understanding before enrolling in the study.

5. One of the survey questions for patients "Do you find it difficult to remember to take oral drugs regularly "--almost all times kids are given meds by parents/caregiver [except may be adolescents] so question does not apply to whole patient group.

6. Kids undergoing cancer therapy commonly have NG/OG tube and the authors have not separated concerns with NG tube administration vs oral administration. I hope they can exclude the patients on getting drug via NG approach or NG+oral approach. For example, Taste/smell is not an issue with NG approach.

7. Some of the survey questionnaire is difficult to comprehend--For example--

Does your child have difficulties swallowing tablets or capsules at time of data collection?

Yes, definitely

Yes, partially

Neutral

Rather not

Not at all

What is the interpretation of "rather not" in response?

Response choices should be like

Yes, always

Yes. most times

Yes, occasionally

No, never.

there are other questions with similar confusing answer choices. Example,

Did your child refuse to take oral drugs in the past?

Yes, definitely

Yes, partially

Neutral

Rather not

Not at all

8. The authors frequently refer AMBORA trial data in the discussion/result section. This could be trimmed and does not add to the study findings. Other authors can compare how adult responses differ vs Pediatric.

9. How the study findings can be used to improve patient and disease related outcomes needs to be highlighted.

10. How youngAMBORA care program differ from periodic pharmacist review practiced at most hospitals? For example, it is common practice in the west to have monthly pharmacist assessment with parents/patient about chemo adherence/administration etc.

11. The side effects are naturally vague and can be grouped as it is confusion and adds to the study jargon. For example, Muscle pain, General pain, Fatigue can be grouped. Similarly urinary side effects--

12. Authors should highlight limitations of mixed-method research in the discussion section.

6. PLOS authors have the option to publish the peer review history of their article (what does this mean?). If published, this will include your full peer review and any attached files.

Reviewer #1: **Yes: **Tanuj Chawla

Reviewer #2: No

---

## [Author Response · Author response to Decision Letter 0]

18 Oct 2024

The response to reviewers has been uploaded as a separate file.

---

## [Decision Letter · Decision Letter 1]

21 Nov 2024

Medication safety with oral antitumour therapeutics in paediatrics (youngAMBORA): A mixed-methods approach towards a tailored care program

PONE-D-24-17810R1

Dear Dr. Metzler,

We’re pleased to inform you that your manuscript has been judged scientifically suitable for publication and will be formally accepted for publication once it meets all outstanding technical requirements.

Kind regards,

Uwe Konerding

Academic Editor

PLOS ONE

Additional Editor Comments (optional):

Reviewers' comments:

Reviewer's Responses to Questions

**Comments to the Author**

1. If the authors have adequately addressed your comments raised in a previous round of review and you feel that this manuscript is now acceptable for publication, you may indicate that here to bypass the “Comments to the Author” section, enter your conflict of interest statement in the “Confidential to Editor” section, and submit your "Accept" recommendation.

Reviewer #1: All comments have been addressed

Reviewer #2: All comments have been addressed

2. Is the manuscript technically sound, and do the data support the conclusions?

Reviewer #1: Yes

Reviewer #2: Yes

3. Has the statistical analysis been performed appropriately and rigorously? 

Reviewer #1: Yes

Reviewer #2: Yes

4. Have the authors made all data underlying the findings in their manuscript fully available?

Reviewer #1: Yes

Reviewer #2: Yes

5. Is the manuscript presented in an intelligible fashion and written in standard English?

Reviewer #1: Yes

Reviewer #2: Yes

6. Review Comments to the Author

Reviewer #1: Looks better now after revisions. The authors have done a retrospective data analysis followed by a prospective survey. This has made their basis for a future study

Reviewer #2: Thank you for addressing all the queries satisfactorily. The manuscript highlights important aspects of methodology and discusses limitations comprehensively.

7. PLOS authors have the option to publish the peer review history of their article (what does this mean?). If published, this will include your full peer review and any attached files.

Reviewer #1: No

Reviewer #2: **Yes: **Ravi M Shah

---

## [Editor Report · Acceptance letter]

26 Nov 2024

PONE-D-24-17810R1 

PLOS ONE

Dear Dr. Metzler, 

I'm pleased to inform you that your manuscript has been deemed suitable for publication in PLOS ONE. Congratulations! Your manuscript is now being handed over to our production team.

Kind regards, 

on behalf of

Dr. Uwe Konerding 

Academic Editor

PLOS ONE